# Exploring *Eimeria* Genomes to Understand Population Biology: Recent Progress and Future Opportunities

**DOI:** 10.3390/genes11091103

**Published:** 2020-09-21

**Authors:** Damer P. Blake, Kate Worthing, Mark C. Jenkins

**Affiliations:** 1Pathobiology and Population Sciences, Royal Veterinary College, Hawkshead Lane, North Mymms AL9 7TA, UK; 2Animal Parasitic Diseases Laboratory, Building 1040, Agricultural Research Service, USDA, Beltsville, MD 20705, USA; kate.worthing@usda.gov (K.W.); mark.jenkins@usda.gov (M.C.J.)

**Keywords:** *Eimeria*, genome, genetics, population structure, chickens

## Abstract

*Eimeria*, protozoan parasites from the phylum Apicomplexa, can cause the enteric disease coccidiosis in all farmed animals. Coccidiosis is commonly considered to be most significant in poultry; due in part to the vast number of chickens produced in the World each year, their short generation time, and the narrow profit margins associated with their production. Control of *Eimeria* has long been dominated by routine chemoprophylaxis, but has been supplemented or replaced by live parasite vaccination in a minority of production sectors. However, public and legislative demands for reduced drug use in food production is now driving dramatic change, replacing reliance on relatively indiscriminate anticoccidial drugs with vaccines that are *Eimeria* species-, and in some examples, strain-specific. Unfortunately, the consequences of deleterious selection on *Eimeria* population structure and genome evolution incurred by exposure to anticoccidial drugs or vaccines are unclear. Genome sequence assemblies were published in 2014 for all seven *Eimeria* species that infect chickens, stimulating the first population genetics studies for these economically important parasites. Here, we review current knowledge of eimerian genomes and highlight challenges posed by the discovery of new, genetically cryptic *Eimeria* operational taxonomic units (OTUs) circulating in chicken populations. As sequencing technologies evolve understanding of eimerian genomes will improve, with notable utility for studies of *Eimeria* biology, diversity and opportunities for control.

## 1. Introduction

*Eimeria* are protozoan parasites that generally invade and develop within epithelial cells of the intestinal tract. Although over 1500 species have been described that infect a wide range of vertebrate hosts including mammals, birds, fish, reptiles and amphibians [1], *Eimeria* are, with few exceptions, absolutely host-specific [2]. For instance, *Eimeria falciformis* will only infect mice and not rats, which can be parasitized by *Eimeria nieschulzi* [3]. *Eimeria* are members of a large group of protozoa that include the parasites *Toxoplasma gondii* and the *Plasmodium* species in the phylum Apicomplexa. Protozoa in this phylum share a conoid structure in the apical complex that plays a role in host cell invasion [4]. *Eimeria* and *T. gondii* are placed in the order Eucococcia, suborder Eimeriina. Although the term ‘coccidiosis’ can be used to describe any disease caused by coccidian parasites, it is generally used to refer to infection by *Eimeria*, and particularly those of poultry. Coccidiosis of chickens is a devastating intestinal disease that inflicts costs in excess of US$ 13.6 billion/UK£ 10.4 billion on the global poultry production industry through the costs of control and losses in production, including egg layers, broilers reared for meat and breeding stock [5]. Avian coccidiosis has been, and commonly continues to be, controlled by routine medication of feed with ionophore drugs or synthetic chemicals. Alternative means of preventing coccidiosis, such as vaccination with low doses of either virulent or attenuated *Eimeria* oocysts, have gained favor for various reasons including repeated appearance of drug-resistance in response to continuous drug use and public demand for reduced drug use in food production [6]. Coccidiosis is often characterized by veterinarians as the most important disease of poultry because of morbidity and mortality associated with *Eimeria* infection [7]. Understanding the molecular epidemiology of coccidiosis is important if one is to protect the greater than 68 billion broiler and 7 billion egg layer chickens that are produced each year worldwide [8].

The disease coccidiosis can be said to begin after the invasion of epithelial cells by sporozoites that are released from sporocysts by the action of bile salts and enzymes, such as trypsin. Sporocysts themselves are contained within oocysts that have a wall composed of glycoprotein and lipid, which is broken down either by the grinding action of the ventriculus (gizzard) in avians or the effect of carbon dioxide and transit through the acidic environment of the stomach (non-avians) [9]. After host cell invasion, sporozoites initiate 2–4 rounds of asexual replication termed schizogony, producing large numbers of merozoites that invade other cells leading to a subsequent round of schizogonous development or to sexual replication with the production of male and female gametes (microgamonts and macrogamonts). Microgamonts develop into microgametes that fertilize macrogametes to form zygotes, subsequently developing into oocysts. Aside from a diploid zygote stage, all other *Eimeria* developmental stages are haploid; a feature that is directly relevant to studies of *Eimeria* genomes and genetics [10,11]. Oocysts are generally released in feces and under the right conditions of temperature, humidity and oxygen content, undergo rounds of meiosis and mitosis (sporulation) to produce haploid sporozoites [12]. Oocysts are extremely resistant to a wide range of environmental conditions that are typically found in commercial broiler houses. In fact, studies have shown that *Eimeria* oocysts remain viable in litter for several weeks between different sets of broilers, infecting the next batch of chickens and thus perpetuating disease indefinitely [13,14]. Most *Eimeria* lifecycles are relatively quick compared to other apicomplexans, commonly represented by pre-patent periods shorter than one week [15], and lack long-term persistence within the host in the absence of an in vivo cyst phase, suggesting an “hit and run” lifecycle. However, it is notable that *Eimeria* species that infect migratory avian species such as crane (*E. gruis* and *E. reichenowi*) or corncrake (*E. crecis* and *E. nenei*) can cause longer lasting disseminated visceral coccidiosis [16,17] and appear to be phylogenetically distinct from the more classical *Eimeria* species that infect mammals and non-migratory avians [18]. Wider sampling of *Eimeria* genomes will contribute to resolution of taxonomy within the genus and relationships with other closely related genera such as *Cyclospora* and *Cystoisospora* [19,20].

There now is consensus that seven *Eimeria* species are capable of infecting the domestic chicken, *Gallus gallus domesticus* [21]. These seven species can be divided into two groups—those causing hemorrhagic disease (*E. brunetti*, *E. necatrix*, and *E. tenella*) and those primarily causing malabsorption (*E. acervulina*, *E. maxima*, *E. mitis*, and *E. praecox*) [10]. Of interest is that each of these seven *Eimeria* species has a predilection for a particular region of the chicken gut. For instance, *E. acervulina* is predominantly found in the duodenum, *E. maxima* in the jejunum and ileum, and *E. tenella* in the caeca [15]. Many species of *Eimeria* cannot reliably be differentiated by microscopy because of similar morphologies and overlapping sizes of the oocyst stage. Molecular techniques such as PCR have proven useful to distinguish different *Eimeria* species. Gene targets include ribosomal genes including the 18S rDNA, internal transcriber spacer (ITS) sequences 1 and 2, as well as genes contained within the mitochondrial genome such as cytochrome c oxidase subunit I (COI) [22,23]. Molecular techniques have contributed to the resolution of taxonomic controversies such as the existence of *E. mivati*, where the identification of two different versions of the 18S rDNA sequence in *E. mitis*, one of which was identical to that purported to be derived from *E. mivati*, proved to be decisive [21]. More intriguingly, others using PCR to amplify and sequence a fragment from the ITS2 region have provided evidence for the existence of three additional *Eimeria* ‘genotypes’, currently referred to as cryptic species or operational taxonomic units OTUx, OTUy, and OTUz [24,25]. Preliminary phylogenetic inference suggests that each cryptic *Eimeria* OTU is related to, but distinct from, recognized *Eimeria* species with close affinities to *E. maxima*, *E. brunetti* and *E. mitis*, respectively [26]. Until recently the cryptic OTU genotypes appeared to be geographically restricted, primarily within the southern hemisphere [25], although recent studies have suggested a wider spatial occurrence [27]. Attempts to resolve questions around the heritage of these and other OTU *Eimeria* genotypes will benefit from recent and future advances in genomics and genetics.

## 2. *Eimeria* Genomes

Eimerian genomes are thought to consist of a nuclear genome including 14 chromosomes of 1–7 Mb [28,29], a mitochondrial genome of ~6200 bp [26] and an ~35 kb circular apicoplast genome [30]. Additional double stranded RNA viral genomes have been described from many *Eimeria* species including *E. brunetti*, *E. maxima*, *E. necatrix* and *E. tenella*, representing a subgenus of the Totiviridae family recently proposed to be named *Eimeriaviruses* [31,32].

Karyotyping of *Eimeria* genomes lagged behind many other eukaryotes, in part because visualization and analysis of cell contents were hampered by the extreme mechanical resistance of the oocyst wall, the breakdown of which typically necessitated such mechanical force that cell contents were disrupted. Pulsed-field gel electrophoresis (PFGE) using chromosomal grade genomic DNA extracted from agarose preserved *E. tenella* sporozoites by Shirley [28] indicated that the organism had at least 14 chromosomes, ranging in size from 1 to 6–7 Mb; observations that were supported by linkage mapping [33] and later by del Cacho, Pages, Gallego, Monteagudo and Sánchez-Acedo [29] via electron microscopy. Del Cacho, et al. [34] developed a less vigorous method of cell wall breakdown using clorhidric (hydrochloric) acid-ethanol and freeze-thawing; this allowed the visualization of intact pachytene synaptonemal complexes during meiosis by transmission electron microscopy and identification of at least 14 chromosomes in *E. tenella* [29]. Interestingly, recent application of third generation Oxford Nanopore genome sequencing to the closely related coccidians *Neospora caninum* and *T. gondii* has reduced their karyotypes from 14 to 13 chromosomes [35,36]. Equivalent studies with *Eimeria* may be similarly informative.

All *Eimeria* genomes sequenced to date feature a segmented chromosome structure characterized by repeat-rich (R) and repeat-poor (P) regions [10,37,38]. The most common repeat sequence is the trinucleotide CAG, which is ubiquitous across the genome and common in protein-coding regions [10,37]. The CAG repeats result in stretches of homopolymeric amino acid repeats (HAARs) that are translated and transcribed, seemingly without affecting protein structure or function [10]. Other repeats such as the heptamer AAACCCT/AGGGTTT are also common, but not within coding sequences where they would disrupt coding frames, as are fragmented retrotransposon-like elements comparable to chromovirus long-terminal repeat (LTR) retrotransposons. High levels of sequence degeneracy suggest that these retrotransposons are unlikely to be functional, although the quality of the sequence assemblies have hindered analysis. While repeat types are well conserved in the sequenced *Eimeria* species, the frequency and location of repeats vary across species [10] and may also vary among strains within species [28]. Shirley [28] observed differing homologous chromosome sizes among different strains of *E. tenella* and postulated that variation in the number of CAG repeats may be one explanation for size variation between strains. No function has been assigned to these repeats and they have not been associated with specific genes or gene families, although it has been suggested that they might contribute to varied levels of recombination and genome evolution [10,37].

To date, all publicly available *Eimeria* genome sequence assemblies exist as scaffolds made up of hundreds or even thousands of contigs or supercontigs (Table 1). The relatively short sequence reads produced by Sanger, Illumina and pyrosequencing have been unable to span many of the repeat-rich regions, precluding assembly to whole chromosome levels. Draft sequence assemblies have been published for all seven of the recognized *Eimeria* species that infect chickens, represented by up to three strains per species [10,39]. A genome sequence of the mouse-associated species *E. falciformis* was released in 2014 [38], followed by the brown rat-associated species *E. nieschulzi* as part of a gametocyte gene discovery project in 2017 [3]. No other *Eimeria* genome sequences have been published to date. Consideration of assembly quality, including the number of contigs and level of curation, led Reid and colleagues to identify three tiers. The *E. tenella* genome was considered to be tier 1, representing the highest quality of assembly and annotation at the time of publication [10]. The *E. falciformis* assembly can also be considered to be tier 1. An intermediate grouping of *E. acervulina*, *E. maxima* and *E. necatrix* was included as tier 2, restricted to automatic post-assembly improvements only. The genome assemblies for *E. mitis*, *E. necatrix*, *E. nieschulzi* and *E. praecox* were least refined, described as or equivalent to tier 3. Recognizing the limitations of these resources, it is striking to note a 1.7-fold difference between the smallest (*E. maxima*, 42.5 Mb) and largest (*E. mitis*, 72.2 Mb; Table 1) assemblies. Comparison with the *T. gondii* genome annotation revealed that 93–99% of the core eukaryotic genes were detectable in the tier 1 and 2 *Eimeria* genomes. As has occurred with *Toxoplasma gondii* and *Neospora caninum* [35,36], the application of third generation sequencing technologies can be expected to provide a dramatic improvement in *Eimeria* genome sequence resources in the near future.

All *Eimeria* genomes sequenced to date have been found to contain two or more paralog clusters of genes that encode surface antigens (SAGs) [38]. Some *E. tenella* SAG proteins have been shown to bind mammalian cells [43] and/or induce inflammatory responses in avian macrophages [44]. A loose association has been suggested between SAG gene number and *Eimeria* species pathogenicity, where species and lifecycle stages that express more SAGs tend towards higher pathogenicity, although a functional link has not been proven [10]. Comparison of *Eimeria* species that infect mammalian or avian hosts suggest distinct clades of SAG genes, with one exception within the SAGa cluster. Protein structural analyses suggest that the equivalent *srs* surface antigen-coding genes within the *T. gondii* derive from a distinct evolutionary origin [10].

In addition to the nuclear genome, apicomplexans have an ~35 kb circular molecule of extrachromosomal DNA located within an organelle termed the apicoplast [45,46]. Thought to have evolved via the symbiotic engulfment of an algal cell, the apicoplast contributes to key cellular functions such as fatty acid production and heme synthesis [47,48,49]. The genome of the *E. tenella* apicoplast was first sequenced by long range overlapping PCR, confirming a comparable gene set to those described for the *T. gondii* and *Plasmodium falciparum* apicoplast genomes [30], and later showing a high degree of similarity to the apicoplast genome of *C. cayetanensis* [50].

Complete mitochondrial genome sequences have also been published for all *Eimeria* species that infect chickens, as well as several species from turkeys and rabbits, supplemented by *E. falciformis* (mouse) and *E. zuernii* (cattle) [26,51,52]. Primarily used for phylogenetic inference, the mitochondrial genomes of *Eimeria* form concatemers of ~6200 bp (6148–6261 bp). The mitochondrial genome sequences contain more than 60% adenine and thymine nucleobases (A + T), and include the protein coding genes cytochrome c oxidase subunits I (COI) and III (COIII), and cytochrome b (CytB). The mitochondrial genomes also encode ribosomal large and small subunit rRNA sequences.

## 3. Comparative Genomics

Coccidian parasites of the family Eimeriidae have traditionally been characterized by features such as sporulated oocyst morphology, including the number of sporocysts per oocyst, and the number of sporozoites per sporocyst [53]. Sporulated oocysts of *Eimeria* species can usually be differentiated from other coccidia by the presence of four sporocysts, each containing two sporozoites. Recent phylogenetic inference using apicoplast and mitochondrial genomes has revealed a close relatedness between the genus *Eimeria* and *Cyclospora* [50], with *C. cayetanensis* located within an eimerian clade (Figure 1). Such phylogenetic comparison has revealed the apparently paraphyletic nature of the genus, although markers such as COI have proven to be equivocal, especially when analysis is restricted to short fragments [54]. Sporulated *Cyclospora* oocysts differ from *Eimeria*, defined by two sporocysts rather than four, but *Eimeria* lines defined by a heritable bisporocytic oocyst appearance have been described following drug selection [55]. Comparative analysis of the *C. cayetanensis* and *E. tenella* genomes suggest that these parasites share common coccidia-like metabolism and invasion pathways, but present with distinct surface antigens, despite detection of a small number of putative *Eimeria* TA4-like SAG coding sequences in the *C. cayetanensis* genome [19]. The functional similarities between *Eimeria* and *Cyclospora* genomes indicate opportunities to use *Eimeria* as model organisms for human pathogens such as *C. cayetanensis* that cannot currently be cultured in vitro or ex vivo.

In addition to *Eimeria* and *Cyclospora*, genome sequence assemblies have been published for five other coccidian genera/species: *Cystoisospora suis* (one strain available in ToxoDB), *Hammondia hammondi* (one), *Neospora caninum* (one), *Sarcocystis neurona* (two) and *T. gondii* (18), all within the family Sarcocystidae. Comparison of the *Eimeria* genomes with members of the Sarcocystidae suggest the absence of synteny, with different repeat occurrence and distributions [10]. This differs from the complete synteny that exists between the mitochondrial and apicoplast genomes of *E. tenella* and *C. cayetanensis* [50]. Microneme and rhoptry organelles are conserved between these genera, although the protein repertoires vary. For example, while many microneme proteins share some structural conservation between *Eimeria* and *T. gondii*, *Eimeria* genomes contain fewer rhoptry kinase (*ropk*) genes [10,57].

Significant opportunities exist for comparison between different *Eimeria* genomes. For example, genome sequencing from phylogenetically distinct species such as *E. gruis* and *E. reichenowi* from migratory crane [18] may inform on phenotypes associated with disseminated visceral coccidiosis, possibly revealing new genera. Similarly, accessing genomes of atypical *Eimeria* species such as *E. leuckarti*, *E. macusaniensis*, *E. truncata* and *E. stiedai*, from horses, alpacas, geese and rabbits, respectively [58,59,60,61], may offer insights into their unique biologies.

## 4. Population Genetics

*Eimeria* field populations are complex. At least seven species can infect chickens, frequently co-occurring in overlapping multi-species infections [62]. The outcome of infection is influenced by the host and the parasite. For example, individual commercial or inbred chickens have been shown to present varied levels of resistance, tolerance or susceptibility to infection by different *Eimeria* species and strains [63,64,65], creating significant polymorphism within and between flocks. Similarly, *Eimeria* strain-specific immunity has been described following infection with *E. acervulina* [66], *E. mitis* [67], *E. maxima* [63] or *E. tenella* [68], indicating the occurrence of strain-specific antigenic polymorphism and varied strain-specific immune selection during subsequent parasite infections. Other notable variables include farmed chicken population structures, where large numbers of chickens are commonly reared at high stocking densities over very short periods, encouraging rapid *Eimeria* cycling. Short parasite generation times provide opportunities for rapid evolutionary events, for example development of differing levels of fecundity or pathogenicity [69,70]. Signatures of selection within *Eimeria* populations of chickens are expected to be diverse, including purifying selection under the influence anticoccidial prophylaxis using drugs, or balancing selection imposed by immunity following natural infection. The use of live parasite anticoccidial vaccines may result in mixed signatures of selection, depending on the nature and diversity of local field populations.

Few population genetics studies have been published for *Eimeria*, in part due to the paucity of appropriate tools. Genome-wide genetic markers have been developed using tools such as Random Amplification of Polymorphic DNA–Polymerase Chain Reaction (RAPD–PCR) and Amplified Fragment Length Polymorphism (AFLP) (reviewed elsewhere [23]), but their utility with complex populations has been limited. Most genetics-led studies have focused on markers derived from the 18S-ITS1-5.8S-ITS2 rDNA repeat, located on chromosome 12 in the *E. tenella* genome [25,71], or markers such as COI in the mitochondrial genome [51]. Primarily used to assess *Eimeria* species occurrence (e.g., [72]), these marker sequences have also been used to define the occurrence of genetically distinct genotypes in different geographic locations or production systems, and to assess association with disease or performance outcomes [73]. In one of the most detailed studies, 248 ITS sequences were produced from *Eimeria* samples collected from 17 countries across Africa, Asia, Europe, North and South America [25]. Here, calculation of Wright’s Fixation Index indicated allopatric diversity for *E. tenella*, but not *E. acervulina* or *E. mitis*, suggesting different population structures for different *Eimeria* species. Possible explanatory factors included the lower fecundity and longer prepatent period of the former species, providing a greater opportunity for genetic isolation. Another feature of the study was the detection of cryptic *Eimeria* OTU genotypes (named x, y and z), previously thought to be restricted to Australia, across much of the southern hemisphere [24,25], although the study was not able to identify their origin. Other locus specific approaches applied to *Eimeria* have included sequencing of multiple cloned PCR amplicons representing loci encoding the vaccine candidates Apical Membrane Antigen 1 (AMA1) and Immune Mapped Protein 1 (IMP1). Targeted sequencing from *E. tenella* revealed low levels of nucleotide diversity with few non-synonymous substitutions and low levels of balancing selection [41,74]. Greater genetic distances were detected between parts of Asia and North Africa, although diversity remained low. Importantly, while these studies have been informative, single locus marker groups leave the majority of the nuclear and the apicoplast genomes unrepresented and risk bias in identification towards the most common genotype(s) in the sampled populations.

In 2015, a new, medium throughput tool was developed for *E. tenella* populations using Sequenom MassARRAY to genotype 55 single nucleotide polymorphisms (SNPs) in two multiplexes [41]. The SNPs were spread across the *E. tenella* karyotype to permit assessment of segregation and recombination, as well as diversity. Using *E. tenella* samples from India, Egypt, Libya and Nigeria, 52 of the SNPs were found to be informative. Analysis of the SNP profiles indicated considerable genome-wide genetic diversity with significant evidence of allopatric evolution. Detection of significant linkage disequilibrium (LD) in North Africa and northern India, but not Nigeria and southern India, suggested different population structures. Subsequent studies suggested contributions from climate and parasite prevalence led to the differences observed in LD, although further study is required [75]. The Sequenom MassARRAY SNP panel was also used to demonstrate that polyclonal infection was common in field populations, and that cross-fertilization occurs at a high frequency during co-infection [41]. Recognizing that Sequenom MassARRAY SNP typing is difficult to develop for routine application in regional laboratories, a subset of 11 SNPs used in the panel were redeveloped as PCR-Restriction Fragment Length Polymorphism (PCR-RFLP) genetic markers [76]. Application of the PCR-RFLP panel to new *E. tenella* samples collected in the United Kingdom and Ireland revealed a tightly restricted haplotype structure that was distinct from haplotypes detected previously in Africa and Asia.

Next generation sequencing (NGS) technologies have supported the development of new approaches to define population structures for a range of micro-organisms. Deep sequencing of PCR amplicons produced using generic or specific primer pairs has been used widely to define the presence/absence (α diversity) and level of occurrence (β diversity) of bacterial populations [77]. The same approach has been developed to define parasite populations for nematodes, termed the ‘nemabiome’ [78], and *Theileria parva* [79]. Three groups have published studies of next generation deep amplicon sequencing for whole *Eimeria* populations. Amplicon sequencing of 18S rDNA was first applied to communities of *Eimeria* from the Australian brush-tailed rock-wallaby [80]. A subsequent study involving *Eimeria* from chickens also focused on the 18S rDNA, using Illumina MiSeq NGS to sequence amplicons produced from commercial and indigenous chickens sampled in India [81]. Sequence analysis provided a sensitive assessment of *Eimeria* species occurrence, validated by standard and quantitative species-specific PCR, and successfully detected the cryptic *Eimeria* genotypes OTUs x and y as well as a range of other related protozoan pathogens. However, detection of low levels of DNA representing other *Eimeria* species not classically associated with chickens did appear to indicate some background noise. Annotation of sequences assigned previously to *Eimeria* infecting rock partridges (*Alectoris graeca*) and wild turkeys (*Meleagris gallopavo*) may have represented farm-level contamination with non-replicating *Eimeria* oocysts or DNA, or false positives [81]. These latter results indicate a requirement for additional validation and/or more specific primers. The second poultry study also targeted the *Eimeria* 18S rDNA, supplemented by COI, but followed a nested PCR approach using inner and outer primers, followed by MiniSeq NGS, in an attempt to increase specificity and sensitivity [27]. Application of the nested 18S and COI NGS assays to live anticoccidial vaccines and field samples collected from backyard and commercial chickens revealed best results for the 18S rDNA, confirmed by quantitative TaqMan PCR. Analysis of the 18S rDNA and COI datasets identified the presence of the cryptic *Eimeria* OTUs x, y and z in North America for the first time [27]. However, as for the publication by Hinsu et al. [81], a series of additional sequences associated with non-chicken hosts were also detected including ferrets, rodents and rock partridge. Furthermore, novel sequences lacking a matching annotated reference sequence were also detected. Considered by the authors to represent up to nine new *Eimeria* OTUs (labelled 1–4 and A–E), further validation is required to confirm that these sequences do not simply represent environmental contamination with non-replicating *Eimeria* oocysts or DNA from other sources, especially given the use of backyard chickens for most of the samples. Combined, these studies demonstrate the value of NGS amplicon sequencing to studies of *Eimeria* populations, but highlight the requirement for optimal primer design and quality control. As *Eimeria* genome sequence assemblies improve, population profiling through whole genome metagenomics may become feasible. Metagenomic approaches for population profiling have been applied to bacterial populations for several years [82], but have not yet become common for parasite communities due in part to expense, given the far greater cost of sequencing larger parasite genomes at an appropriate depth. The fragmented nature of many parasite genomes has also been limiting.

## 5. Challenges

It is now more than 100 years since the significance of coccidiosis was first recognized in poultry [83,84], but many notable gaps persist in our knowledge. We are farming more poultry than ever before, with a greater reliance on poultry meat and eggs for provision of dietary protein for human consumption [85]. Concurrently, we are attempting to reduce routine reliance on relatively indiscriminate, genus-wide, anticoccidial drugs, replacing them with precise species- and sometimes strain-specific vaccines. The appearance of cryptic *Eimeria* genotypes is problematic. While these genotypes are likely to be controlled by anticoccidial drugs to the same extent as the recognized species, they may be capable of escape from commercial anticoccidial vaccines [86]. Understanding genetic diversity, population structure and capacity to evolve has never been more important for control of *Eimeria* species parasites. Key questions surround the ability of *Eimeria* to evolve to escape current live or future subunit vaccines. The nature of eimerian genome evolution is poorly defined, including rates of cross-fertilization in field populations and the occurrence of genetic recombination. Indeed, the possibility of viable hybridization between *Eimeria* species, as has been described for other apicomplexans (e.g., *Plasmodium berghei* and *P. yoelii* [87]), remains unclear. Developing new, high throughput genome-wide genetic markers beyond the current ribosomal and mitochondrial options will be important to understand how *Eimeria* populations behave under deleterious drug or vaccine selection, and to better explore host-parasite cospeciation [54].

## 6. Opportunities

The ongoing revolution in next- and third-generation sequencing technologies offers good prospects for studies with veterinary pathogens. Costs associated with DNA extraction, library preparation and sequencing are falling fast. Other restraints including stringent requirements for high nucleic acid template quality and quantity, previously limiting for organisms that cannot be cultured in vitro, are relaxing. Routine laboratory or even field sequencing using platforms such as Oxford Nanopore is now improving accessibility to genomic sciences. For *Eimeria*, applications to genomes, comparative genomics and population genetics can be numerous. Long read sequencing can be used to improve the current draft genome sequence assemblies, confirm karyotype numbers and access spatially, temporally and phenotypically distinct isolates. Work can expand to include more *Eimeria* species, including those from other livestock and wild animal populations, to explore synteny and ancestral associations. In time, sequencing may be applied to routine diagnostics, differentiating vaccine strains from field isolates and informing selection of optimal chemoprophylactic programmes. In the short term, identification of improved primers for NGS deep amplicon sequencing from *Eimeria* populations and strategies to control for the influence of background/non-target DNA is likely to prompt a much wider uptake of routine sequencing studies.

## 7. Conclusions

As methods to control *Eimeria* are evolving so are the tools we use to investigate them. Detection of unexpected genome- and population-level complexities has renewed interest in study of these parasites that can support improved controls and safeguard the sustainability of livestock production.

## Figures and Tables

**Figure 1 genes-11-01103-f001:**
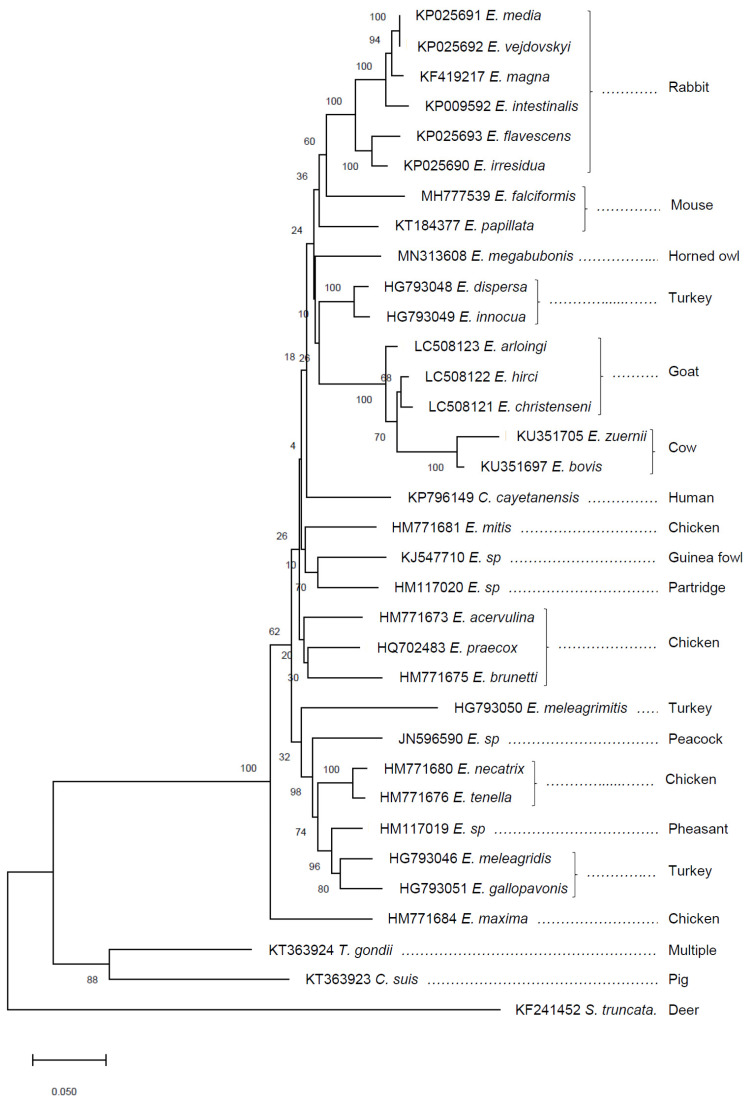
Neighbor joining phylogeny of partial coccidian mitochondrial cytochrome c oxidase subunit I (COI) sequences. The optimal tree is shown with the percentage of replicate trees in which the associated taxa clustered together in the bootstrap test (1000 replicates) indicated next to the branches. GenBank accession numbers are shown for each sequence, followed by the parasite species and host (common name) identity. Sequences were aligned using CLC Main Workbench (version 8.0.1), creating a 788 bp alignment. The phylogeny was inferred using MEGA X [56], with the evolutionary distances computed using the Kimura 2-parameter method.

**Table 1 genes-11-01103-t001:** A summary of genome sequence assemblies available for Eimeriidae family parasites. Data derived from ToxoDB ([40]; accessed on 14 July 2020) and references as cited.

Parasite Species	Parasite Strain	SequencinG Platform	Assembly Size (Mb)	No. Contigs/Supercontigs	Reference
*E. acervulina*	Houghton	Illumina HiSeq 2000	45.8	3415	[10]
*E. brunetti*	Houghton	Illumina HiSeq 2000	66.9	8575	[10]
*E. maxima*	Houghton	Sanger capillary and Roche GS-FLX 454	46.0	22,259	[39]
	Weybridge	Illumina HiSeq 2000	42.5	3564	[10]
*E. mitis*	Houghton	Illumina HiSeq 2000	72.2	15,978	[10]
*E. necatrix*	Houghton	Illumina HiSeq 2000	55.0	3707	[10]
*E. praecox*	Houghton	Illumina HiSeq 2000	60.1	21,348	[10]
*E. tenella*	Houghton	Sanger capillary and Illumina GAIIx	51.9	4664	[10]
	Nippon-2	Illumina GAIIx	na	na	[10,41]
	Wisconsin	Illumina GAIIx	na	na	[10,41]
*E. falciformis*	Bayer Haberkorn 1970	Illumina GAIIx	43.7	753	[38]
*E. nieschulzi*	Landers	Illumina HiSeq 2000	63.0	33,467	[3]
*C. cayetanensis*	CHN_HEN01	Roche GS-FLX 454, Illumina GAIIx and Illumina HiSeq 2500	46.8	4811	[19]
	NF1_C8	Illumina MiSeq	44.4	738	[42]

na = not applicable (reads aligned against the *E. tenella* Houghton reference strain, not de novo assembled).

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
