# Peer review of "Exploring Eimeria Genomes to Understand Population Biology: Recent Progress and Future Opportunities"

_genes, 2020, doi:10.3390/genes11091103_

Round 1
Reviewer 1 Report
The review paper "Exploring Eimeria genomes to understand population biology: recent progress and future opportunities" is a well written review of progress on Eimeria research in the NGS era. I found this paper to be a really enjoyable read that provided an overview of genomic research in Eimeria with a particular focus on species and strains infecting Gallus gallus. The manuscript highlighted important advances and problematic areas that still need work in the sequencing of Eimeria genomes while posing future avenues for meaningful research that can have direct application to disease control. Aside from reviewing for minor typos (line 316 ...climate and parasite prevalence led(?) to the differences...) I recommend this review paper for publication.
Author Response
We thank Reviewer 1 for their positive review. We have corrected the error on line 316 (please refer to line 319 in the revised manuscript) and checked for additional errors.
Reviewer 2 Report
The review “Exploring Eimeria genomes to understand population biology: recent progress and future opportunities” summarizes in an excellent and comprehensive way existing knowledge on the genomes of Eimeria spp. This review might be also of interest for researches focussing on Eimeria-related protozoan parasites.
I have only a very few and minor comments:
Line 130: Do you mean "Hydrochloric acid" ?
Line 189: What kind of "association"? Please be more specific.
Line 242: I think there is also a Besnoitia genome assembly which may need to be mentioned. But mabe I am wrong.
Author Response
We thank Reviewer 2 for their positive review. In response to their comments:
Line 130. Yes, this is correct. The authors (del Cacho et al) used this term in the original work, which we intended to reflect. In response, we have included both versions to avoid confusion.
Line 189. Text has been added to expand on this point. Lines 190-191.
Line 242. Yes, there is at least one Besnoitia genome sequence assembly (Schares et al. (2017) Genome Announcements), although it does not appear to be available in resources such as ToxoDB. We haven’t mentioned Besnoitia or other members of the family Sarcocystidae in this table, instead choosing to focus on the Eimeriidae. We could expand the table, at the editor’s discretion, but believe these details will be covered by others in this Special Edition.
Reviewer 3 Report
This is a particularly well-written and timely review on Eimeria genomics and population genetics by experts in the field. It is a worthwhile summary of knowledge.
I have little to add as comments, which are minor:
- the first 70 lines or so provide fundamental basic background knowledge on the biology of Eimeria species. Most of this is well known and could be deleted if space is an issue. I personally would prefer to see a more modern and relevant introduction to the topic. However I leave it to the authors to decide.
- The phylogenetic tree in Fig. 1 is a token effort to represent the relationship between Eimeria and Cyclospora. It is missing a number of species that would be in the ingroup and the choice of Toxoplasma as the outgroup is questionable on its own (perhaps include Isopora?). I appreciate the authors are not suggesting this is a detailed tree - why not simply create a simple graphic to represent the relationships (and then maybe include more eimeria species). The fact that Eimeria is paraphyletic is clearly worth mentioning.
-
Line 334 the reference (Vermeulen et al., 334 2016) is incorrectly formatted –should be a number?
-
Line 339 OTUs x and y , what are x and y?
- A paper published earlier this year suggested many of the markers used to study Eimeria population genetics in the rodent were inappropriate. This is a decent paper in the field and supports the authors points: https://onlinelibrary.wiley.com/doi/full/10.1002/ece3.5992
Author Response
We thank Reviewer 3 for their positive review. In response to their comments:
- Many thanks for this comment – this is something that we thought about during preparation of the manuscript. We recognise that the beginning of the paper provides a fundamental background to Eimeria that may be unnecessary for many readers, but believe that it is helpful to set the scene for those that may be less familiar. We have retained this section, at the editor’s discretion.
- Agreed – the intention of the figure was to illustrate the relationship between Eimeria and Cyclospora. We have followed this reviewers advice and expanded the figure to include more species in the in- and outgroups.
- Thanks for pointing this out. It is strange that EndNote left this reference in a different format – an error that we have now resolved.
- Good question! OTUs x and y are cryptic Eimeria genotypes that remain enigmatic. We have introduced them in the introduction (please refer to the end of section 1), and now added reference to them in section #4.
- Many thanks for highlighting this paper – it certainly does support our point. We have now included reference to this paper.